# Job Stress and Mental Well-Being among Working Men and Women in Europe: The Mediating Role of Social Support

**DOI:** 10.3390/ijerph18052494

**Published:** 2021-03-03

**Authors:** Aziz Mensah

**Affiliations:** 1Bielefeld Graduate School in History and Sociology (BGHS), Bielefeld University, 33615 Bielefeld, Germany; aziz.mensah@uni-bielefeld.de; Tel.: +46-737-0195-28; 2School of Health, Care, and Social Welfare, Mälardalen University, 722 20 Västerås, Sweden

**Keywords:** job stress, mental well-being, social support, gender, working adults, Europe

## Abstract

Job stress is one of the most common health issues in many organizations, particularly among women. Moreover, an increase in job stress with low social support may have an adverse effect on mental well-being. This study investigated the mediating role of social support in the relationship between job stress and mental well-being among working men and women in Europe. A cross sectional data set from the 2015 6th European Working Conditions Survey on 14,603 men and 15,486 women from 35 countries in Europe was analyzed. The study applied Hayes process macro 4 modelling technique to estimate the direct, indirect, and total effects of job stress on mental well-being among working adults. The study further used the Hayes process macro 59 model to estimate the gender difference in the mediating effect. The results showed that job stress had a direct negative effect on mental well-being among workers in Europe (β=−0.2352,p<0.05). However, there were significant gender differences in the relationship (β=−0.3729,p<0.05), with women having higher effect size than men (men: β=−3.9129,p<0.05 vs. women: β=−4.2858,p<0.05). Furthermore, the indirect effect showed that social support mediated the relationship of job stress on mental well-being (β=−0.0181, CI: −0.0212−0.0153). Nevertheless, the mediating effect of social support did not differ among men and women. This study provides evidence that job stress has a negative impact on mental well-being among working adults, and social support mediates this relationship. The results highlight the importance of the role of support from colleagues and supervisors at the work place, which may help reduce job stress, and improve mental well-being. Sociological and occupational health researchers should not ignore the role of gender when studying work environment and jobs in general.

## 1. Introduction

A growing body of research on occupational behavior and health has identified job stress to be one of the most common health issues in many organizations in Europe [1] and globally [2,3]. For instance, in Europe, the 4th European working condition survey conducted in 2005 revealed that more than 40 million (22%) of working adults were affected by job stress [3]. Also, a study conducted in 2014 showed that 77% of the US population experience job stress [4]. A nationwide survey in Japan also indicated that more than half of the working population were affected by job distress (i.e., stress, anxiety, and worry) on a daily basis [2]. Moreover, the increasing number of women in the labor market has created much attention for the need to fully understand the potential gender differences in job stress in organizations [5]. In fact, 2018 report from the European Commission indicated that men accounted for 78.1% of the labor force in Europe, while women accounted for 66.6% [6]. Despite the significant proportion of women in the labor force, the labor market is clearly segregated by gender in most countries, which means that women and men work in different sectors and occupations. Gender segregation affects the psychosocial work environment among men and women, and contributes to gender inequality in job stress [7].

### 1.1. Job Stress Theoretical Approaches

Job stress has gained much attention because it is one of the main determinants of mental well-being among working adults [8]. Job stress occurs when the demand placed on an individual at the work place exceeds their perceived ability to successfully cope with the situation, resulting in a harmful reaction [9]. Much literature has identified four highly influential theoretical approaches of job stress. The first theoretical approach was developed by Karasek [10]. He posits that the exposure to a stressful situation may be accounted for by a combination of two key dimensions, known as the job demand and job control. Job demand is operationalized in terms of the task requirement and it includes workloads, time pressure, and role conflict. The second dimension—job control—is the extent to which a person may have control over his work activities. Job control consists of decision latitude and skill discretion. A person is considered as having low job control when they do not have autonomy over their work, and are also denied the opportunity to use their skills [10]. The Job Demand-Control (JDC) theory states that the most stressful situation occurs when there is high demand and low control [11].

Based on their empirical research, Johnson and Hall [12], and Johnson et al. [13] extended the Job Demand Control model to include a third dimension known as work place social support. The extension was later recognized by Karasek and Theorell [11]. Work place social support may be defined as the degree of helpful social interaction that is obtained from both supervisors and colleagues at the work place [11]. Work place social support comprises colleague support and supervisor support. Supervisors structure work environment and provide useful feedback and information to workers. According to Wayne and colleagues [14], the social interaction between workers and supervisors may determine the attitude and behavior of employees in the work environment. Colleague support also offers useful resources to workers when they listen, offer assistance, and enhance team cohesion [15]. The Job Demand Control Support (JDCS) model provides distinction for social support in terms of isolated jobs and collective jobs [12]. According to the JDCS model, the most stressful work situation occurs when there is high job demand, and low control and social support at the workplace [11]. However, this model has been criticized as a ‘male model’ because its impact on negative health outcomes are more pronounced among men than women [12].

The second theoretical approach of job stress is the Effort Reward Imbalance (ERI), which was developed by Siegrist [16]. The author defined the ERI as when the demand made on an employee lack the reciprocity of fairness in terms of the reward that is received. The demand includes work hours, physical and emotional load, time pressure, and frequency of interruption. Reward received also includes job security, salary or wages, and career opportunities. Siegrist [16] established that high demand or effort with low reward will increase exposure to job stress. Moreover, Colquitt [17] indicated that the ERI theoretical model may generally be concerned with organizational injustice, which deals with inequities in employees conduct in the work environment. According to the author, employees who experience organizational injustice may have higher levels of job stress.

The third theoretical approach of job stress is the Job Demand Resources model that was developed by Demerouti and colleagues [18] to understand job stress and burnout. The authors posit that the range of job demand and job resources that was examined in the JDCS and ERI model was too superficial to appraise the complexity of job characteristics and job process. Here, job characteristics were divided into job demand and job resources. Job demand was defined as “those physical, social, organizational aspects of the job that require sustained physical or mental effort and are therefore associated with certain physiological and psychological costs” [18]. For example, work overloads, insecurity, and conflict. On the other hand, job resources are factors that are available for an individual to cope with the job demand, and this includes social support, autonomy, performance feedback; career opportunities, job security, salary, and role clarity. An individual with excessive job demand faces stress when they exhaust their job resources [18,19].

The fourth theoretical approach is the transactional process model that was developed by Lazarus and Folkman [20]. Lazarus and Folkman defined job stress as “a particular relationship between a person and the environment that is appraised by the person as taxing or exceeding his or her resources and endangering his or her well-being” [20]. This model primarily focusses on the continuous interaction and adjustment between a person and his environment. The relationship between the person and the environment consist of two major phases known as the cognitive appraisal and coping [20]. While the cognitive appraisal assesses whether the demand that is placed on the individual threatens his well-being, coping on the other hand refers to the cognitive and behavioral effort that the individual takes to help reduce the stress. For example, mobilizing help or support from supervisors and colleagues.

Although the theoretical approaches of job stress that have been discussed so far have been based on several questionnaire items of measure; nevertheless, few studies have based job stress on the single-item measure [21,22]. The use of single-item questionnaires to measure job stress has become popular because of it psychometric and non-psychometric benefits to researchers and practitioners. Regarding the psychometric benefits, Elo et al. [22] demonstrated that single-item measure of job stress is a satisfactory operationalization, and a valid construct that can be used in work-life to replace the theoretical models with several questionnaire items. In addition, Gilbert and Kelloway [21] indicated that the single-item measure of job stress is more robust because it allow a respondent to personally consider the relevant components of his or her own facet rather than draw the attention of the respondent to some predetermined component of a construct. In terms of the non-psychometric benefits, Gilbert and Kelloway [21] argued that using a single-item to measure job stress is cost effective and takes less time for the respondents to complete the survey. Moreover, because this measure of job stress is easy to use, it is possible for researchers and employers to frequently monitor the level of stress at work in order to take immediate action. In view of these benefits, this study relied on the single-item measure of job stress as measured in the European Working Condition Survey 2015.

### 1.2. Relationship between Job Stress and Mental Well-Being

Many studies have suggested that job stress may be associated with negative health and mental well-being outcomes, including blood pressure, musculoskeletal disorders, cardiovascular disease, anxiety, depression, burnout, emotional exhaustion, dissatisfaction, and poor mental health [10,11,12,13,18,22]. For example, a longitudinal study conducted by Tyssen et al. [23] in Norway found evidence that job stress is a risk factor of poor mental well-being among working adults. A systematic review on both cross-sectional and longitudinal cohort study that was conducted recently indicated a strong association between job stress and poor mental health [24]. Although the relationship between job stress and negative mental well-being is well established, some scholars have argued that there is a gender difference in the relationship [5,25]. According to these scholars, the relationship between job stress and adverse health outcomes among men and women may differ because of their exposure to different job stress levels. They argue that men and women differ in terms of the jobs they do, how they are perceived and treated in the society, and kind of working conditions available to them. Using the gender role theory, the authors attributed the gender difference in the relationship to the concentration of men and women into different sectors (horizontal segregation), occupational distribution (vertical segregation), and double burden role in work and family life [5,26]. For instance, in explaining horizontal segregation, the authors indicated that while men are concentrated in certain sectors of employment such as industry and manufacturing, women are mostly concentrated in teaching, nursing, clerical, and sales jobs, which are highly related to job stress. In regards to vertical segregation, the authors explain that men and women may differ in their hierarchical areas and levels, and that women continue to occupy more precarious jobs, occupy less prestigious positions, and earn less wages as compared to men. Meanwhile, evidence suggests that these kinds of working conditions may be highly related to job stress [7]. Also, using the double burden role to explain the gender difference in the relationship between job stress and mental well-being, many scholars have argued that while women have increased their engagement in work activities, women continue to engage more in household and care responsibility than men [27,28]. Moreover, the double burden role may put additional stress on women which may subsequently influence their mental health outcomes [7].

However, there have been inconsistent results in the gender difference in the relationship between job stress and negative mental well-being outcomes [29,30,31], with some showing higher association between job stress and negative mental well-being among men [31], while others show higher association between job stress and negative mental well-being among women [25,29].

### 1.3. Mediating Role of Social Support

Most prior studies that investigated the relationship between job stress, social support, and mental well-being suggested that social support may buffer or moderate the effect of job stress on both physical and psychological well-being, indicating that social support could weaken or reduce the adverse effect of job stress on mental health [32,33]. However, according to Mackinnon [34] and Frazier et al. [35], the results from these studies only examined whether the independent variable (job stress) and the dependent variable (mental well-being) have the same relations across groups (social support), and rather ignored the process or the pathways through which job stress influence mental well-being. For this reason, there has been a growing interest in the mediating role of social support in the relationship between job stress and negative mental well-being in sociological and occupational health research [36,37,38].

According to the mediational effect or model, job stress may have a direct effect on social support, and social support may in turn predict mental well-being. For instance, some scholars have argued that there is an inverse relationship between stress and social support, and that those who have lower levels of job stress, perceive their social support to be high [37]. Meanwhile, numerous studies have also suggested that social support is an important factor in maintaining good physical and psychological health [33,39,40]. More importantly, higher levels of social support may serve as a protective factor against negative mental well-being such as depression, anxiety, poor life satisfaction, and poor quality of life [36].

The mediation hypothesis suggest that social support may mediate the relationship between job stress and mental well-being outcomes [36,37,38]. For example, a cross-sectional study conducted among young Chinese nurses in 16 tertiary hospitals in Chengdu found evidence that suggest that social support mediates the relationship between job stress and negative mental well-being such as depression, anxiety, and job burnout [36]. Similarly, a study conducted by Wu and colleagues [37] among 1464 banking staff in China concluded that the relationship between job stress and cynicism is mediated through social support. Contrarily, a systematic review analysis of Viswesvaran et al. [38] on relevant studies did not find evidence that social support may mediate the relationship between job stress and mental well-being. Thus, findings from the mediational hypothesis have been inconsistent. Meanwhile, little is known whether the mediating role of social support in the relationship between job stress and mental well-being among working adults may vary with gender, as no attention has been focused on this question.

To avoid any potential bias in the current study due to other possibility of the direction of the effect between job stress and mental well-being, it will be interesting to explore other alternative mediational model of social support, where social support mediates the relationship of mental well-being on job stress. To test this model, the present study needs to find out if those with negative mental well-being predict low social support, and low social support in turn predict job stress.

Based on the existing theories, the following research hypotheses were formulated:

**Hypothesis** **1** **(H1).**
*Job stress is negatively associated with mental well-being among working adults.*


**Hypothesis** **2** **(H2).**
*Job stress is more negatively associated with mental well-being among women than men.*


**Hypothesis** **3** **(H3).**
*Job Stress is negatively associated with social support among working adults.*


**Hypothesis** **4** **(H4).**
*Social support is positively associated with mental well-being among working adults.*


**Hypothesis** **5** **(H5).**
*Social support mediates the relationship between job stress on mental well-being.*


**Hypothesis** **6** **(H6).**
*Social support has a higher mediating effect in the relationship between job stress on mental well-being among women than men.*


**Hypothesis** **7** **(H7).**
*Social support mediates the relationship between mental well-being on job stress.*


In view of the above discussion, it is clear that few studies have been conducted on the pathways through which job stress is linked to mental well-being [36]. To the best of our knowledge, this is the first study to be conducted with gender variation and with a comprehensive country sample in Europe. Thus, the primary objective of this study is to examine whether the mediating role of social support in the relationship between job stress and mental well-being among working adults in Europe varies with gender. The study further explored the potential mediating effect of social support in the relationship of mental well-being on job stress.

The proposed theoretical framework of the relationship between job stress, social support, and mental well-being is displayed in Figure 1 and Figure 2.

## 2. Materials and Methods

### 2.1. Data

The data for this study was from the 6th wave of the European Working Conditions Survey (EWCS), which was collected by the European Foundation for the Improvement of Living and Working Conditions in 2015. The survey covered about 44,000 employees in 35 countries in Europe. This included all the 28 countries in the European Union (EU), Norway, Switzerland, Albania, Serbia, North Macedonia, Montenegro, and Turkey. The 2015 EWCS employed a multi-stage stratified sampling to select employees who are 15 years and above for a face-to-face interview. Further details on sampling techniques and data collection process are described elsewhere [41,42]. This study focused on employees who are between 16–64 years, non-disabled, non-retired, and not full-time students.

### 2.2. Measures

#### 2.2.1. Mental Well-Being

Mental Well-Being was measured with the WHO-5 Questionnaire Index. WHO-5 well-being index is known to be very effective in the evaluation of emotional well-being and depression [43,44]. It consists of 5 items inquiring about respondents’ feelings over the last two weeks. The five items (“I have felt cheerful and in good spirits?”, “I have felt calm and relaxed?”, I have felt active and vigorous?”, “I woke up feeling fresh and rested?”, “My daily life has been filled with things that interest me?”) were rated on a 1 (all of the time) to 6 (at no time) point Likert scale. The rating was recoded in an ascending order where higher levels were assigned with higher values, while lower levels were assigned with lower values. The internal consistency of reliability estimated was very good with a Cronbach α level of 0.88. The scale has also been validated in different fields and with different health outcomes [45]. The sum score theoretically ranges from 0 to 25. But it is recommended to multiply the score by 4 to translate it into percentage scale from 0 to 100, so that higher score represent higher well-being, and lower score represent lower well-being [45].

#### 2.2.2. Social Support

Social support at work explains the extent to which workers experience collaboration and support from their supervisors and colleagues. A short version of social support scale was used as the measurement of social support in this study [46,47]. This variable was measured with two items: “Your manager helps and supports you?” and “Your colleagues help and support you?”. Response options from both supervisor and colleague support ranged from 1 (always) to 5 (never). The items were further recorded in an ascending order so that higher scores referred to higher social support.

#### 2.2.3. Job Stress

Job stress was evaluated as a single-item measure as contained in the EWCS 2015 survey. The single-item measure of job stress is a short version of measuring job stress and helps researchers to reduce assessment burden. The variable was measured with the question: “Do you experience stress in your work?”. Responses were rated from the range of 1 (always) to 5 (never) points on a Likert scale. The rating was recoded in an ascending order, where higher levels were assigned with higher values, while lower levels were assigned with lower values. The scale was validated with different health outcomes [45].

#### 2.2.4. Covariates

Demographic variables, socio-economic positions, and working characteristics were also controlled for in the analysis. Demographic variables were age, gender (men and women), marital status (married or cohabiting and single or widowed), and living with child (yes or no). Socio-economic positions were measured with education (International Standard Classification of Education—2011) and occupation (International Standard Classification of Occupation—2008). Education was categorized into four groups (primary school or less, secondary, post-secondary, and tertiary). Variables from working characteristics also included shift work, fixed working time, the standard industrial classification (NACE), and working hours. Shift work was measured with the question: “Do you work shift?”. The response from this question was categorized into “yes” or “no”. Fixed time was also measured with the question: “Do you work fixed starting and finishing times?” The response option was “yes” or “no”. NACE was further classified into four groups (agricultural, industry, service, and others). The study also controlled for countries. Thirty-five countries were selected for this research, namely, Denmark, Sweden, Finland, Norway, Germany, France, Netherland, Belgium, Austria, Luxembourg, Switzerland, Ireland, United Kingdom, Spain, Italy, Greece, Portugal, Malta, Cyprus, Turkey, Croatia, Estonia, Bulgaria, Lithuania, Slovakia, Slovenia, Hungary, Romania, Czech Republic, Latvia, Poland, Montenegro, Serbia, Albania, and North Macedonia.

### 2.3. Statistical Analysis

Descriptive statistics were performed to report the mean, standard deviation, frequency, and percentage of all the variables. Three different analytical strategies were used in this study. First, the study estimated the bivariate relationship between measured variables using Spearman rank correlation. The Spearman rank correlation is best known in estimating the correlation between ordinal variables [48]. Second, Hayes [49] process macro in the SPSS version 22.0 was used to estimate the direct effects, indirect effects, and the moderated mediation effects. The model can be used to simultaneously estimate the relationship of the measured variables [49]. In regards to the mediation test, the Hayes process macro model 4 was employed to estimate the mediating effect [49]. There are different steps involved in testing for the mediating effect.

Firstly, the study estimated path *a*, which is the direct effect of the predictor (job stress or mental well-being) on the mediator (social support). Secondly, the present study estimated path *b*, which is the direct effect of the mediator (social support) on the outcome variable (job stress or mental well-being). Thirdly, the study quantified the product of path *a* and *b* (*ab*) to obtain the indirect effect. Lastly, path *c*, which is the direct effect of the predictor variable (job stress or mental well-being) on the outcome variable (mental well-being or job stress) was also estimated. Although it is assumed that the standard normal distribution for estimating the *p*-values of the indirect effects are normally distributed, this can only happen when there are large sample sizes [50]. Therefore, this study considered a nonparametric resampling method known as a bootstrapping procedure, which does not assume normality of the sampling distribution. The bootstrapping procedure helps us to obtain accurate indirect effects, and is less sensitive with small sample size [50,51]. In view of this, the study applied a 5000-sample bootstrapping procedure to estimate the bias-corrected 95% confidence interval (CI) to determine the significance of the indirect effect. According to Hayes [49], if the upper boundary and the lower boundary of the bias-corrected 95% CI do not contain zero, then the indirect effect is significant. Additionally, this study estimated the total effect by adding the indirect effect and the direct effect of the independent variable on the dependent variable [50]. The use of effect in this study is in it technical sense and does not by any means imply causation [52]. For ease of interpretation and comparison, the path coefficients were standardized [53].

The third analytical strategy that was employed in this research was the Hayes index of moderated mediation model, which was outlined as model 59 by Hayes [49]. This method enabled us to examine whether the mediating effect of social support in the relationship between job stress on mental well-being is moderated by gender. To test for this, the study estimated whether path *a* or path *b*, or both path *ab* are moderated by gender [49,54]. The model also enabled us to test the gender difference in the direct effect of job stress on mental well-being. 5000-bootstrapping procedure was adopted to estimate the 95% bias-corrected CI in order to observe the index of moderated mediation. The moderation mediation or the index of moderation is significant if the bias-corrected 95% CI’s do not include zero [49]. The descriptive statistics and the Spearman rank correlation analysis was stratified by gender. While this study used Stata V14 [55] to perform data preparation and the descriptive statistics, SPSS version 22.0 was also employed to perform the analytical strategy.

## 3. Results

### 3.1. Socio-Demographic Characteristics

Table 1 provides details of the general descriptive statistics of workers from the 6th EWCS 2015. While 48.53% of the sample were men, women represented 51.47%. The average age of working men (41.43 ± 11.60 years) was very similar to women (41.92 ± 11.27 years). This study observed that most workers in Europe had obtained secondary education (men = 58.51% vs. women = 49.52%), followed by tertiary education (men = 31.08% vs. women = 40.31%). Overall, women reported higher levels of education than men in Europe. Furthermore, women were more likely to engage in fixed working time (73.78%) as compared to men (67.42%). Meanwhile, the frequency of engaging in shift work was slightly higher among women (26.33%) than men (25.15%). Also, the average weekly working hour was higher among men (40.70 ± 10.05) than women (35.69 ± 10.57). Although slightly more working men were married or cohabiting (65.83%) than women (63.60%), most men did not live with children (55.66%). Women had higher frequency of living with children (52.11%) than men (44.34%).

Men reported slightly lower job stress as compared to women (men = 2.89 ± 1.16 vs. women = 2.96 ± 1.12). This study also observed higher levels of social support among workers in Europe, although they were quite similar among men and women (men = 4.22 ± 0.89 vs. Women = 4.24 ± 0.91). Meanwhile, men reported averagely higher levels of mental well-being than women (men = 69.70 ± 19.43 vs. women = 67.64 ± 20.16).

### 3.2. Bivariate Analysis

The correlations between variables are presented in Table 2. The results showed a weak and negatively significant correlation between job stress and mental well-being among men and women. More precisely, the correlation for men was $\left(\rho =-0.2412\right)$ and women was $\left(\rho =-0.2480\right)$. Social support at work also decreased job stress. Furthermore, social support at work had positive a correlation with mental well-being. Overall, there were similar patterns of correlation between the measured variables for both men and women.

### 3.3. Direct and Indirect Effects

Hayes Process Macro model 4 was applied to estimate the direct, indirect, and total effects in the relationships between job stress, social support, and mental wellbeing. First, this study estimated the relationship of job stress on mental well-being via social support. The results for the direct and indirect effects are presented in Table 3 and Figure 3. After controlling for demographic variables, working conditions, socio-economic positions, and countries, the study found a direct and negative effect of job stress on mental well-being (β=−0.2352,p<0.05) among working adults. Job stress negatively and directly influenced social support (β=−0.0820,p<0.05). Furthermore, there was a direct positive relationship between social support and mental well-being (β=0.2213,p<0.05). Also, the total effect of job stress via social support on mental well-being among working adults was β=−0.2533,p<0.05. Based on the bias-corrected bootstrapping method that was recommended by Hayes [49], the analysis showed that social support significantly mediated the relationship between job stress and mental well-being. More specifically, the indirect effect of job stress on mental well-being was β=−0.0181, 95% CI: −0.0212–0.0153. A proportional analysis of the indirect effect (mediator) indicated that about 7.1% of the variance in mental well-being was explained by social support.

In regards to the relationship of mental well-being on job stress through social support, the analysis from Table 4 and Figure 4 indicated that mental well-being positively influenced social support (β=0.2407,p<0.05), and social support in turn negatively influenced job stress (β=−0.0217,p<0.05). Based on the recommended bias-corrected bootstrapping by Hayes [49], social support significantly mediated the relationship of mental well-being on job stress (β=−0.0052, 95% CI: −0.0081–0.0024) among working adults. The overall total effect of mental well-being on job stress via social support was β=−0.2471,p<0.05. A proportional analysis conducted indicated that only 2.1% of the variance in mental well-being was explained by social support.

### 3.4. Moderated Mediation Effects

To test whether gender moderated the relationship between job stress and mental well-being via social support among working adults, Hayes process macro model 59 was applied to estimate the moderated mediation modelling as outlined by Hayes [49]. The model allowed us to moderate gender on all direct and indirect paths. Table 5 showed that after adjusting for demographic variables, working conditions, socio-economic characteristics, and countries, gender did not moderate the relationship between job stress and social support. Furthermore, the relationship between social support and mental well-being did not differ by gender. However, there was significant gender difference in the relationship between job stress and mental well-being (β=−0.3729, p<0.05), and that women had higher associations as compared to men (men: β=−3.9129, p<0.05 vs. women: β=−4.2858, p<0.05). The bias-corrected bootstrapping method did not reveal a statistically significant moderating mediating effect $\left(\beta =-0.0397,\;95\%\;\mathrm{CI}:\;-0.1414\;0.0598\right)$. This indicates that gender did not moderate the mediating effect of social support in the relationship between job stress and mental well-being.

## 4. Discussion

This study applied Hayes process macro model to investigate the mutual relationships between job stress, social support, and mental well-being among working adults in Europe with particular focus on gender differences. To the best of our knowledge, this is the first study to examine whether job stress is indirectly related to mental well-being via social support, and whether the direct and indirect effects of job stress on mental well-being are moderated by gender. In addition, the study explores whether social support mediated the relationship of mental well-being on job stress. The findings suggested that job stress had a direct and negative relationship with mental well-being, and that women had higher effects as compared to men. Furthermore, the study confirmed that social support played a significant mediating role in the relationship between job stress and mental well-being, but it did not differ by gender. Also, social support acted as a mediator in the relationship of mental well-being on job stress. These findings contribute to the deeper understanding of stress at work and the need for support from colleagues and supervisors to improve and promote mental health and safety at the work place.

### 4.1. Job Stress, Social Support, and Mental Well-Being

The results from this study are in line with numerous previous findings of a negative association between job stress and mental well-being among workers [10,11,18,23,24]. Possible reasons for these findings have been attributed to higher job demand, low job control, and imbalance between effort and reward that is reported among workers [10,16,18,47,56]. This was observed in the study as job stressors such as irregular work time, higher proportion of shift work, and higher work overtime that was reported in this study was shown to be related to poor health outcomes [57,58]. The findings in this study are particularly important considering the fact that the single-item questionnaire that was used to measure job stress showed satisfactory construct and validity.

In terms of the gender differences in the relationship between job stress and mental well-being, the study revealed that the negative association of job stress on mental well-being were more consistent among women than men, although some studies have reported inconsistent findings [31]. This result is in line with previous studies that suggested that there is a higher association in the relationship between job stress and negative mental well-being among women than men [25,29,59,60]. Several possible reasons may explain these findings. For instance, it was established that working women spend more time on care and household responsibility than men, irrespective of their time spent on work activities [61]. Therefore, they may suffer more from double role burden and role conflict as compared to men [61]. Consequently, the stress from both career and care and household responsibility may subsequently be associated with greater poor mental well-being among women than men [5,30]. The gender difference in the association may also be attributed to vertical segregation [5] since prior studies noted that there are few women in Europe with higher job positions than men, and that women engage in more precarious jobs as compared to men [62]. This was evident in this study as women had fewer management positions and higher elementary job positions as compared to men, although their educational attainment levels were higher than men. At the same time, women engaged in more precarious jobs in terms of shift work and part-time work than men. Another possible reason for the findings may also be due to horizontal segregation in the labor market [5,26]. The analysis revealed that while there is a higher concentration of men in the industry sector than women, there is a higher concentration of women in the service sector than men. Meanwhile, prior studies showed that workers in the service sector may experience higher levels of job stress than the industrial sector [7,63], and this may in turn, have more detrimental mental well-being outcomes among women than men [7]. Also, it is thus plausible that the results may perhaps be due to the fact that women may be more prone to self-report poor mental well-being as compared to men that may be prone to substance abuse [64].

The study confirmed the hypothesis that job stress has a negative and direct effect on social support [37]. This suggests that workers who experience low stress at the work place may maintain high level of social support, while those with high level of job stress may not have enough social support to mobilize. In line with numerous studies that claim that social support may positively influence mental well-being [33,40,65], this study found a positive association of social support on mental well-being among workers in Europe. These results may be explained by the fact that social support may serve as a protective factor in maintaining physical and psychological health [40]. Another possible explanation for this finding is that social support provides a positive effect, stability in life, and self-worth [33] among workers.

Consistent with previous studies [36,37], the mediation analysis provided strong evidence that social support significantly mediated the relationship between job stress and mental well-being among working adults in Europe. These findings may perhaps be explained by the fact that social support at work may promote self-meaningfulness and mental well-being among workers [66]. The positive association of social support on mental well-being and the mediating effect of social support in the relationship between job stress and mental well-being partly confirmed the findings of Viswesvaran and colleagues [38]. Although the authors found a positive impact in the relationship between social support on mental well-being regardless of the level of job stress, social support did not mediate the relationship between job stress and mental well-being.

Regarding the gender difference in the mediational effect of social support, the Hayes process macro model 59 showed that the mediating effect of social support in the relationship between job stress and mental well-being did not differ among men and women. This finding may perhaps be explained by the similar levels of social support that was reported among men and women in Europe, indicating that gender has little impact on the amount of social support that is experienced by a worker [47,67]. Besides, Wang et al. [31] demonstrated that social support at work may be equally important for both men and women in preserving good mental health. Nevertheless, it is perhaps possible that the lack of evidence to support the hypothesis may be due to the small number of items that were used to measure social support among men and women in this study. Previous studies in this field of research with more comprehensive items of social support have indicated that there is gender difference in the experience of social support at the work place [68].

Even though the indirect effect of job stress on mental well-being via social support that was estimated in this research was informative, the study further explored other important potential pathways, where social support acted as a mediator in the relationship of mental well-being on job stress. The results showed that mental well-being had a direct and positive effect on social support, and social support in turn, had a negative and direct effect on job stress. Consequently, the study found a low but significant mediating effect of social support in the relationship of mental well-being on job stress among working adults in Europe. This means that regardless of the direction of the pathways or effects between job stress and mental well-being, social support played a mediating role among working adults in Europe.

### 4.2. Practical Implication

This study has several practical implications that may be useful to occupational health practitioners and policy makers who wish to reduce psychosocial strain as a result of job stress in order to promote good health and productivity at the work place. To reduce stress and increase mental well-being at the work place, government, policy makers, and managers must provide workers with good working conditions including effective support from both colleagues and supervisors, reduce long working hours, and increase career opportunities [69]. Furthermore, effective organizational interventions such as flexible working arrangement, improvement in communication, and job redesign may serve as an effective way of reducing stress at the work place [70].

In addition, government and occupational health practitioners may also promote a culture of recognition to reduce the impact of stress and well-being at the work place [71]. Also, as indicated by the European Research Program Horizon 2020, the study of gender equality and innovation must be promoted in sociological and occupational health research such as job stress and mental well-being in order to improve scientific knowledge and effective production of services that may be suitable for potential markets [5].

There must be promotion and encouragement of work-family practices at the work place for both men and women in order to reduce both job stress and any stress that may emanate from family responsibilities. Finally, managers and organizations must design effective stress management and well-being training programs for their employees on how to handle and reduce negative psychosocial factors at the work place.

### 4.3. Limitations and Strength

This study is subject to some limitations. First, although this study controlled for some confounders, factors from behavioral and biological determinant of health [72,73] were not considered in the theoretical model due to lack of data availability. Nevertheless, the social factors in terms of educational level and social class based on occupation that were included in this study are known to be important social determinant of health outcomes [74]. Second, the result could not infer causality due to the cross-sectional design of the study [75]. Third, self-report measures were used in this study, and these measures may be prone to some biases that might not reflect the true status of workers [76]. Nonetheless, it was established that using self-reporting to measure occupational and mental health outcomes is a highly suitable and recommended evaluation technique, which may be considered to be superior to other techniques [77]. Fourth, prospective study in this field of research often based their measurement of job stress on multiple measurements. However, this study was based on single-item assessment of job stress. Therefore, the findings from this study should be taken with caution since using a single-item measurement to estimate job stress may increase the risk of misclassification [29,78]. That notwithstanding, the single-item measurement that was used in this study helped the respondent to easily and uniformly imagine the meaning of job stress to provide appropriate answers [79]. Finally, only social support at work was used as a mediating factor. However, apart from social support, there may be other potential mediating factors that may link job stress and mental well-being. Notwithstanding the above limitations, the study is the first to investigate the relationship between job stress and mental well-being via social support among working men and women in the whole of Europe.

### 4.4. Future Research

The study recommends that future research on the relationship between job stress, social support, and mental well-being should further be tested on longitudinal design research in order to determine causality. Further, it is important to consider other mediating variables that may link the relationship between job stress and mental health outcomes in future studies. For instance, cognitive flexibility, optimism, and resilience are factors that may influence the relationship between stress at the work place and mental well-being [80]. Since the research was based on the single-item approach of job stress, future studies should be conducted with other theoretical approaches of job stress such as job demand control, effort reward imbalance, job demand resource, and the transactional process model to have a broader and a more dynamic understanding of the issues under discussion. Finally, and considering the fact that socio-economic policies that exist between countries differ in Europe, it might be worthwhile for future studies to focus on exploring the cross-country variations in the mediating role of social support in Europe, particularly among men and women. This could broaden the knowledge on social policies across countries and the importance of its role in promoting gender equality and influencing job stress levels, social support, and mental well-being.

## 5. Conclusions

In general, job stress had significantly negative and direct effect on mental well-being among working adults, but the magnitude of effect was higher among women than men.

Furthermore, this study observed that although social support mediated the relationship of job stress on mental well-being among working adults, there was no gender difference in the mediating effect. Also, social support mediated the relationship between mental well-being on job stress. The present study highlights the importance of the role of gender in sociological and occupational health research. Therefore, governments, organizations, and policy makers should develop and implement work-family friendly policies that may promote gender equality and further improve employment and working conditions for men and women. There is also the need for organizations to train their employees to fully understand and adequately meet the support needs of workers.

## Figures and Tables

**Figure 1 ijerph-18-02494-f001:**
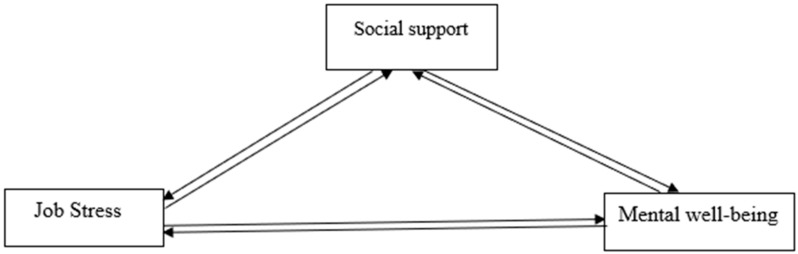
Conceptual framework of the study.

**Figure 2 ijerph-18-02494-f002:**
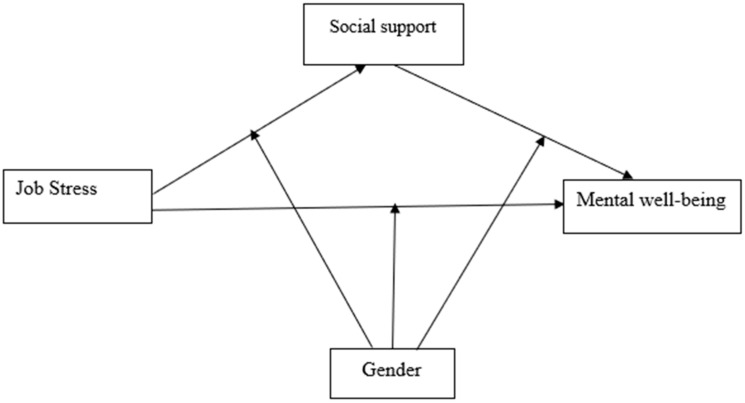
Conceptual framework of the study.

**Figure 3 ijerph-18-02494-f003:**
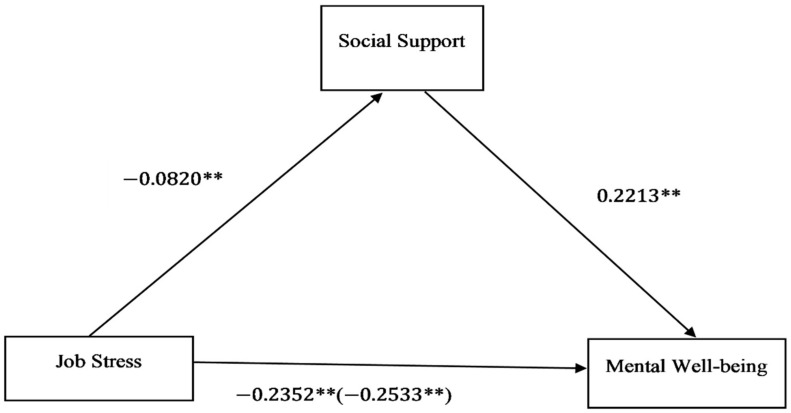
Direct, indirect, and total effects from the Hayes Process Macro model 4. All standardized coefficients are adjusted for demographic characteristics, socio-economic positions, working characteristics, and countries. Total effect is in parenthesis. Significance level: **p<0.05.

**Figure 4 ijerph-18-02494-f004:**
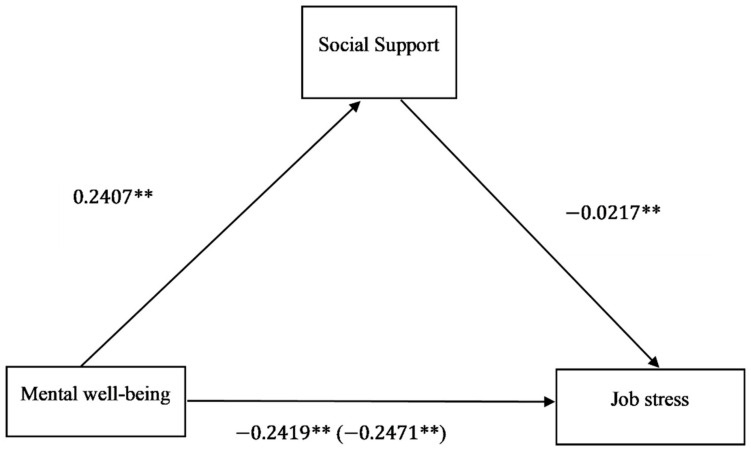
Direct, indirect, and total effects from the Hayes Process Macro model 4. All standardized coefficients are adjusted for demographic characteristics, socio-economic positions, working characteristics, and countries. Total effect is in parenthesis. Significance level: **p<0.05.

**Table 1 ijerph-18-02494-t001:** Descriptive statistics of working men and women from the 6th European Working Condition Survey 2015.

Variables	Men		Women	
	*n* = 14,603	% or Mean ± SD	*n* = 15,486	% or Mean ± SD
Age		41.43 ± 11.60		41.92 ± 11.27
Marital Status				
Single or Widowed	4990	34.17%	5637	36.40%
Married or Cohabiting	9613	65.83%	9849	63.60%
Living with Child				
No	8128	55.66%	7416	47.89%
Yes	6475	44.34%	8070	52.11%
Education				
Primary School or Less	555	3.80%	376	2.43%
Secondary	8544	58.51%	7669	49.52%
Post-Secondary	966	6.62%	1198	7.74%
Tertiary	4538	31.08%	6243	40.31%
Occupation				
Armed Forces Occupation	134	0.92%	14	0.09%
Managers	848	5.81%	636	4.11%
Professionals	2245	15.37%	3845	24.83%
Technicians and Associates	1822	12.48%	2086	13.47%
Clerical Support Workers	1060	7.26%	2163	13.97%
Sales and Service Workers	2228	15.26%	4145	26.77%
Agricultural Workers	204	1.40%	80	0.52%
Craft and Related Trades	2949	20.19%	515	3.33%
Machine Operators	1854	12.70%	473	3.05%
Elementary Occupation	1259	8.62%	1529	9.87%
NACE				
Agricultural	452	3.10%	184	1.19%
Industry	4911	33.63%	2035	13.14%
Service	8668	59.36%	12,332	79.63%
Other	572	3.92%	935	6.04%
Weekly Hour		40.70 ± 10.05		35.69 ± 10.57
Fixed Work Time				
Yes	9846	67.42%	11,426	73.78%
No	4757	32.58%	4060	26.22%
Shift Work				
Yes	3672	25.15%	4078	26.33%
No	10,931	74.85%	11,408	73.67%
Job Stress		2.89 ± 1.16		2.96 ± 1.12
Social Support		4.22 ± 0.89		4.24 ± 0.91
Mental Well-being		69.70 ± 19.43		67.64 ± 20.16

Notes: SD is the Standard Deviation. *n* is the Sample size. NACE is the Industry Standard Classification System.

**Table 2 ijerph-18-02494-t002:** Correlation among measured variables by gender.

Variable	Job Stress	Social Support	Mental Well-Being
Men			
Job stress	1		
Social support	−0.0847 **	1	
Mental Well-being	−0.2412 **	0.2782 **	1
Women			
Job stress	1		
Social support	−0.1002 **	1	
Mental Well-being	−0.2480 **	0.2727 **	1

Notes: Significance level: **p<0.05.

**Table 3 ijerph-18-02494-t003:** Effects from the Hayes Process Macro model 4 on the mediating effect of social support in the relationship between job stress and mental well-being.

Variable	Effects	SS	MWB
JS	Direct	−0.0820 **	−0.2352 **
SS	Direct		0.2213 **
SS	Indirect		−0.0181 (−0.0212–0.0153)
	Total		−0.2533 **

Notes: Significance level: **p<0.05. (): Confidence Interval. JS: job stress; SS: social support; MWB: mental well-being.

**Table 4 ijerph-18-02494-t004:** Effects from the Hayes Process Macro model 4 on the mediating effect of social support in the relationship between mental well-being on job stress.

Variable	Effects	SS	JS
MWB	Direct	0.2407 **	−0.2419 **
SS	Direct		−0.0217 **
SS	Indirect		−0.0052 (−0.0081–0.0024)
	Total		−0.2471 **

Notes: Significance level: **p<0.05. (): Confidence Interval. JS: job stress; SS: social support; MWB: mental well-being.

**Table 5 ijerph-18-02494-t005:** Effects from the Hayes Process Macro model 59 on the moderated mediating effect of social support in the relationship between mental well-being and job stress among men and women.

Variable	Effects	SS	MWB
JS * Gender	Direct	−0.0090	−0.3729 **
SS * Gender	Direct		0.0651
	Index of moderation mediation		−0.0397 (−0.1414 0.0598)

Notes: Significance level: **p<0.05. (): Confidence Interval. JS: job stress; SS: social support; MWB: mental well-being.

## Data Availability

The original data for this research was collected from the European Foundation for the Improvement of Living and Working Conditions. Details of the study design, data collection process, and characteristics of measured variables can be obtained from the homepage https://www.eurofound.europa.eu/surveys/european-working-conditions-surveys/sixth-european-working-conditions-survey-2015 (accessed on 15 July 2020).

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
