# Peer review of "Job Stress and Mental Well-Being among Working Men and Women in Europe: The Mediating Role of Social Support"

_ijerph, 2021, doi:10.3390/ijerph18052494_

Round 1
Reviewer 1 Report
Abstract:
The study confirms the results obtained previously by other researchers, rather than providing evidence. The author himself notes in the first sentence in "introduction": "A growing body of research on occupational behavior and health have identified job stress to be one of the most common health issues in many organizations in Europe"
Most studies on stress reveal gender differences in the level of perceived stress - this is nothing new.
Introduction:
"Gender segregation affects the psychosocial work environment among men and women, and contributes to gender ine quality in job stress [5]." - The thesis is not supported by research results, but only by the opinion of the cited authors.
"In view of these benefits, our study relied on the single-item measure of job stress as measured in the European Working Condition Survey 2015. Line 126." The author of the article is one person, but many times present the research as a team of authors. (e.g. we, our).
"Meanwhile, little is known whether the intermediary role of social support in the relationship between job stress and mental well-being among working adults may vary with gender, as no attention has been focused on this question." - There is a lot of such research!
The theoretical part is understandable and the stress theories mentioned are well described. The author understands and knows the basic theories of organizational stress.
Materials and Methods
The author refers to research that was carried out 5 years there. During this time these data became obsolete. Additionally, they are no longer available on the website.
3. Results
The author's strength is the ability to conduct advanced statistical analyzes. Unfortunately, the obtained data do not contribute much to the understanding of stress in employees.
Reviewer 2 Report
a) Good potential article, but not in its present form.
b) 7 Research Hypothesis are too many.
c) Data from 2015 are old: lot of issues have happened during the last 6 years in terms of job conditions and specially in job gender conditions. Additional comparison with new data are needed.
d) Minor aspects in format: single lines at the end of a page are better placed in another new page, like this example:
3.2. Bivariate Analysis (339 line)
e) Articles like this one are needed: Kamaldeep Bhui:
BJPsych Bull. 2016 Dec; 40(6): 318–325. doi: 10.1192/pb.bp.115.050823 PMCID: PMC5353523 PMID: 28377811Perceptions of work stress causes and effective interventions in employees working in public, private and non-governmental organisations: a qualitative study
Kamaldeep Bhui,1 Sokratis Dinos,2 Magdalena Galant-Miecznikowska,1 Bertine de Jongh,1 and Stephen Stansfeld1
f) References are not quoted from A to Z and are confusingAuthor Response
Please see the attachment.
Thank you.

Round 2
Reviewer 1 Report
Thank you for your time and effort to improve this article. The introduced changes significantly improved the quality of the article. I wish you good luck in conducting research and publishing the results.
Reviewer 2 Report
Thanks for the better now adequate article format.
It has greatly improved.